# Difference in TMPRSS2 usage by Delta and Omicron variants of SARS-CoV-2: Implication for a sudden increase among children

**Sosuke Kakee**[1,2], **Kyosuke Kanai**[1], **Akeno Tsuneki-Tokunaga**[1], **Keisuke Okuno**[2], **Noriyuki Namba**[2], **Katsuyuki Tomita**[3], **Hiroki Chikumi**[4], **Seiji Kageyama**[1]*

1 Division of Virology, Department of Microbiology and Immunology, Faculty of Medicine, Tottori University, Yonago, Japan, 2 Division of Pediatrics and Perinatology, Department of Multidisciplinary Internal Medicine, Faculty of Medicine, Tottori University, Yonago, Japan, 3 Department of Respiratory Medicine, National Hospital Organization Yonago Medical Center, Yonago, Japan, 4 Department of Infectious Diseases, Yonago, Japan

* skageyama@tottori-u.ac.jp

## Abstract

It has been postulated from a combination of evidence that a sudden increase in COVID-19 cases among pediatric patients after onset of the Omicron wave was attributed to a reduced requirement for TMPRSS2-mediated entry in pediatric airways with lower expression levels of TMPRSS2. Epidemic strains were isolated from the indigenous population in an area, and the levels of TMPRSS2 required for Delta and Omicron variants were assessed. As a result, Delta variants proliferated fully in cultures of TMPRSS2-positive Vero cells but not in TMPRSS2-negative Vero cell culture (350-fold, Delta vs 9.6-fold, Omicron). There was no obvious age-dependent selection of Omicron strains affected by the TMPRSS2 (9.6-fold, Adults vs. 12-fold, Children). A phylogenetic tree was generated and Blast searches (up to 100 references) for the spread of strains in the study area showed that each strain had almost identical homology (>99.5%) with foreign isolates, although indigenous strains had obvious differences from each other. This suggested that the differences had been present abroad for a long period. Therefore, the lower requirement for TMPRSS2 by Omicron strains might be applicable to epidemic strains globally. In conclusion, the property of TMPRSS2-independent cleavage makes Omicron proliferate with ease and allows epidemics among children with fewer TMPRSS2 on epithelial surfaces of the respiratory organs.

## Introduction

Since COVID-19 was recognized at the end of the year 2019 [1], it spread globally within several months and became a pandemic [2]. Although the pandemic continued with the periodic emergence of new variants, it has become endemic in 2023. Indeed, the World Health Organization (WHO) has reported that SARS-CoV-2 infections decreased in five of the six WHO regions (-14% to -76%) except for the Western Pacific Region (+14%) (COVID-19 Epidemiological Update- 24 November 2023). Nevertheless, the WHO is still tracking several lineages of

**Funding:** This work was supported in part by a Grant-in-Aid for Scientific Research on Infection Control and Prevention by the International Platform for Dryland Research and Education, Tottori University. The funders had no role in study design, data collection and analysis, decision to publish, or preparation of the manuscript.

**Competing interests:** The authors have declared that no competing interests exist.

Omicron variants and has identified four variants of interest: XBB.1.5, XBB.1.16, EG.5 and BA.2.86. Fortunately, epidemic strains are no longer categorized as variants of concern [3].

The Omicron wave caused a substantial increase in infections among children compared with the Delta wave [4]. A study in Australia reported that 117 pediatric patients were admitted during the Delta wave (1 June 2021–21 December 2021) compared with 737 during the Omicron wave (22 December 2021–30 September 2022), with the peak number of weekly admissions being 14 for Delta and 53 for Omicron [5]. Another report from the UK highlighted the increase in cases in children aged under 1 year old. The proportion of the age group (<1 year) increased steeply to 42% of children admitted to hospital with COVID-19 in the 4-week period (14 December 2021 to 12 January 2022) in which Omicron variants were becoming dominant in the UK according to Coronavirus (COVID-19) latest insights, 30 March 2023 (https://www.ons.gov.uk). The percentage in the Omicron wave was much higher than those in earlier periods: 33% of children (8 months, January to August 2020), 30% (8 months, September 2020 to April 2021), and 30% (8 months, May 2021 to 13 December 2021). Similarly, a US cohort also suggested that the incidence rate of SARS-CoV-2 infection with the Omicron variant was 6 to 8 times that of the Delta variant in children younger than 5 years [6].

There is a potential explanation for the substantial increase among children infected with Omicron variants. Variants prevalent up to the period of the Delta wave were known to make efficient use of transmembrane protease, serine 2 (TMPRSS2) for cell entry. However, amino acid mutations in the spike protein of Omicron variants reduced the efficiency of the TMPRSS2-mediated cell entry and favored the endocytosis pathway, which is a cell entry pathway that does not use TMPRSS2 [7, 8]. It is also known that TMPRSS2 expression in pediatric airway cells is lower than in adults. Therefore, it can be hypothesized that the decreased availability of TMPRSS2 in the Omicron strain led to an increase in the number of pediatric patients [9].

Several host-genetic factors of a gene cluster on chromosome 3 were reported to be involved in the proliferation of SARS-CoV-2 in the human body [10]. The risk-associated DNA segment modulated the expression of several chemokine receptors including CCR5 of SARS-CoV-2 carriers [11]. Other factors, such as inborn errors of type I interferons may also be involved in SARS-CoV-2 proliferation [12]. These findings suggest that it is favorable to carry out the characterization of epidemic SARS-CoV-2 strains spreading among an indigenous population to minimize the virus selection caused by such genetic factors.

In the present study, we isolated Delta and Omicron variants from patients who visited two hospitals within a single city of Japan and assessed the required levels of TMPRSS2 for the proliferation of these Delta and Omicron isolates to examine the cause of the sudden increase in COVID-19 cases among pediatric patients immediately after the onset of the Omicron wave.

## Materials and methods

### Ethical approval

The present study was approved by the Institutional Review Board represented by the Dean of the Faculty of Medicine, Tottori University, Japan (No.20A138, October 30, 2020) and have conducted from November 1, 2020, onwards. All procedures performed in studies involving human participants were in accordance with the 1964 Helsinki declaration and its later amendments or comparable ethical standards.

### Ethical issues including informed consent

Sample collection was carried out after obtaining informed consent from patients through a website notification and/or poster presentations at the out-patient departments in two

hospitals ('opt-out' system). It was guaranteed that the freedom to decline participation as a research subject, and refusal to participate would have no negative effect on the person, and that he/she would be able to continue to receive necessary medical cares. Transparency was preserved to the patients for all information and the information was ready to be discarded according to the patient will. Medical records have been accessed only by selected persons on limited occasions. A guardian was selected to protect human rights for a pediatric patient. Only when all the necessary conditions were fulfilled as described above, patient samples were collected and shipped to the laboratory anonymously.

## Patient samples and virus isolation

Nasopharyngeal swab or saliva samples were collected from SARS-CoV-2 PCR-positive patients who visited Tottori University Hospital or Yonago Medical Center in Yonago city, Tottori prefecture, Japan during the period from July 2021 to August 2022.

The samples in 1 mL UTM media (Cat. 350C, Copan Japan, Kobe city, Japan) were transported to our laboratory and a portion (300 μL) was inoculated into a TMPRSS2-expressing Vero cell line (VeroE6/TMPRSS2 [13], JCRB1819, JCRB Cell Bank, Suita city, Osaka) and incubated at 37°C in a 5% $CO_2$/ 95% air atmosphere for one hour. After removing the residual inoculum by washing the surface of the cells extensively, the cells were maintained in Dulbecco's Modified Eagle Medium supplemented with 5% fetal bovine serum and antibiotics until cytopathic effect was observed. The supernatant was stored at -80°C as viral stock until further analyses.

## Infection in TMPRSS2-positive and TMPRSS2-negative cells

One day before infection, two different Vero cell lines, TMPRSS2-positive and -negative Vero cells (VeroE6 cell line lacking TMPRSS2 expression: Vero76 [14], IFO50410, JCRB Cell Bank), were seeded onto a 6-well plate ($1.0 \times 10^6$ cells/well). Clinical isolates were inoculated ($1.0 \times 10^5$ copies) onto the cells and incubated at 37°C for 1 hour. The cells were washed twice with phosphate-buffered saline to remove inoculated viruses in culture supernatant and maintained in culture medium (2 mL per well) for 72 hours. A portion of the culture supernatant (140 μl) was collected at the indicated time points during the 72 hours culture period and subjected to real-time reverse transcription-polymerase chain reaction (RT–PCR) to determine the viral production levels using RNA copy number in the culture medium.

## RNA extraction, reverse transcription, and polymerase chain reaction

Viral RNA was extracted using QIAamp Viral RNA Mini Kit (Qiagen, Tokyo, Japan) or ISO-SPIN Viral RNA (NIPPON GENE, Tokyo, Japan). RT-PCR and the determination of cDNA sequences were performed using previously reported methods with slight modifications [15]. In brief, the coding regions of SARS-CoV-2 spike protein were amplified by RT-PCR using a One Taq One-Step RT-PCR kit (New England BioLabs, Tokyo, Japan) and a primer pair (35F: 5′–AAGGGGTACTGCTGTTATGT–3′, 41R: 5′–AGCTGGTAATAGTCTGAAGTG–3′) for first round PCR. The first-round PCR products were subjected to a second round PCR with Amplitaq Gold™ 360 Master Mix (Thermo Fisher Scientific, Tokyo, Japan), and M13-tailed primer pairs 35F/35R(5′–TTAATAGGCGTGTGCTTAGA–3′), 36F(5′–TCAGCCTTTTCTTA TGGACC–3′/36R(5′–TCCAAGCTATAACGCAGC–3′), 37F(5′–TTAGAGGTGATGAAGTCA GA–3′)/37R(5′–TGTTCAGCCCCTATTAAACA–3′), 38F(5′–TAACCAGGTTGCTGTTCTT T–3′)/38R(5′–CAATCATTTCATCTGTGAGCA–3′), 39F(5′–CAGATCCATCAAAACCAAGC –3′/39R(5′–GCAAGAAGACTACACCATGA–3′), 40F(5′–TCAGAGCTTCTGCTAATCTTG– 3′)/40R(5′–GTAATTTGACTCCTTTGAGC–3′), and 41F(5′–TTGCCATAGTAATGGTGACA

–3')/41R(AGCTGGTAATAGTCTGAAGTG–3'). For Omicron BA5 variants, a modified primer 37FBA5(5'–TTAGAGGTAATGAAGTCAGC–3') was used instead of the 37F primer in the second round PCR. The second round PCR products were then subjected to the sequencing reactions.

## Determination of nucleotide sequences

Nucleotide sequences were determined using a BigDye Terminators v3.1 Cycle Sequence kit in accordance with the manufacturer's instructions (Thermo Fisher Scientific). Isolated viruses were subtyped using the sequencing results. Variants were classified into two groups according to age (Adults: 16 years or older and Children: 15 years or younger). The key amino acids of spike protein were analyzed in the relation to usage of TMPRSS2 and endocytosis.

## Phylogenetic analysis

The nucleotide sequences of the amplified spike region were determined using a BigDye. Terminator v3.1 Cycle Sequencing kit in accordance with the manufacturer's instructions (Life Technologies). M13 and PCR primers were used for the sequencing reaction. Nucleotide sequences excluding the primer regions (3,822 bases encoding the spike region, 21,563–25,384 of Wuhan-Hu-1) were aligned with sequences obtained from the International Nucleotide Sequence Database. Of 69 sequences, 20 spike sequences were determined partially, but these encompassed the region (22,824–24,152 of Wuhan-Hu-1). Inspection, manual modification, and evolutionary analysis of the sequences were conducted in Molecular Evolutionary Genetics Analysis Version X (MEGA X). A phylogenetic tree was constructed using the neighbor-joining method (1,000 bootstrap replications) in MEGA X. An estimate of the mean evolutionary diversity was also conducted using MEGA X [16]. The lineage of SARS-CoV-2 Omicron variant was determined using the International Nucleotide Sequence Databases (https://www.insdc.org).

## Statistical analysis

Statistical analyses were performed using the Mann-Whitney U test and a *P*-value less than 0.05 was considered statistically significant.

## Results

### Properties of viral isolates

Delta variants were isolated from nine adults, but none were obtained from pediatric patients because of extreme rarity of infections in this population. Omicron variants were isolated from 38 adults and 22 from pediatric patients (Table 1). Delta variants were collected from July to

**Table 1. Characteristics of host and viral isolates.**

|  | Delta | Omicron | |
|---|---|---|---|
|  | Adults | Adults | Children |
| Number of samples* | 9 | 38 | 22 |
| Male/female | 4/5 | 20/18 | 11/11 |
| Averaged age (range) | 31 (25–62) | 55 (20–91) | 6 (0–14) |
| Collection period, year/month/date | 2021/07/14-2021/09/18 | 2022/01/05-2022/07/07 | 2022/01/12-2022/08/13 |
| Key amino acids (%) at 655, 679 and 681** | H (100), N (100), R (100) | Y (100), K (100), H (100) | Y (100), K (100), H (100) |

*: Samples were collected once from each case.

**: Related to S2' cleavage and endosomal entry. Positions are according to Wuhan-Hu-1.

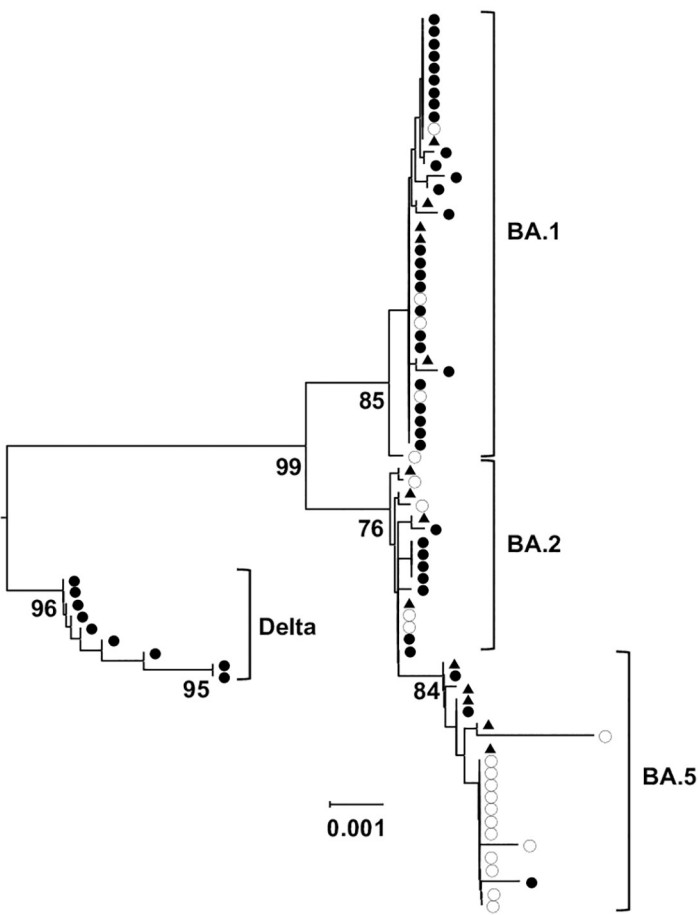

**Fig 1. Phylogenetic tree of SARS-CoV-2 clinical isolates.** The phylogenetic tree was generated from the SARS-CoV-2 spike protein coding region. Isolates from adults (≥20 years, ●) and children (≤14 years, ○) (GenBank accession numbers, LC793397-LC793465) are shown with reference strains (▲). The reference strains with the greatest distances from the present isolates were selected from the result of BLAST searches of 100 beta-coronavirus sequences in the International Nucleotide Sequence Databases. The scale shown the genetic distance. Wuhan-Hu-1 (NC_045512) was used as the outgroup.

September 2021. Omicron variants were collected from adults from January to July 2022, whereas those from pediatric patients were obtained in almost simultaneous period from January to August 2022. Among the key amino acids in the spike protein of SARS-CoV-2 related to the efficiency of S1/S2 and S2' cleavage, P681R was detected in all analyzed Delta variants. As for the Omicron variants, H655Y, N679K, and P681H were detected in all strains analyzed in the present study (Table 1).

Nucleotide sequences of SARS-CoV-2 spike genes of the tested samples (GenBank accession numbers, LC793397-LC793465) were mostly identical to the reference strains retrieved from the International Nucleotide Sequence Database using Betacoronavirus BLAST (>99.63% similarity with 100 references) (Fig 1).

## Different proliferation efficiency of Delta and Omicron variants in TMPRSS2-positive and -negative cells

The amount of progeny viruses of Delta variants in supernatants was significantly higher in cultures of Vero cells in the presence of TMPRSS2 at all sampling times at 24, 48 and 72 hours

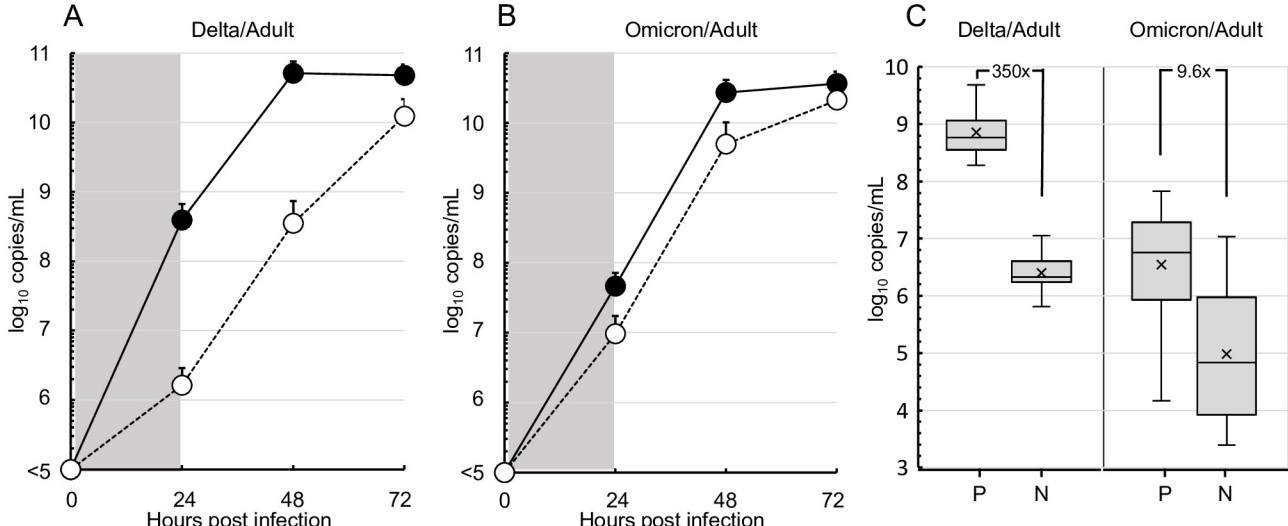

**Fig 2. Proliferation efficiency of the Delta and Omicron variants from adult patients in TMPRSS2-positive or -negative Vero cells.** (A) The production levels of Delta variants (n = 3, GenBank accession numbers, LC793397-LC793399) were determined by SARS-CoV-2 genomic RNA copy number in culture supernatants and compared between TMPRSS2-positive (●) and -negative (○) Vero cell cultures up to 72 hours. Data plots are shown as mean±SD from three independent experiments. (B) The levels of Omicron variants (n = 3, LC793430-LC793432) were also compared similarly. (C) Means and distributions of the production levels of Delta (n = 6, LC793397-LC793400, LC793404, LC793405) and Omicron variants (n = 24, LC793430-LC793442, LC793444-LC793452, LC793455, LC793456) in the supernatants of TMPRSS2-positive (P) or -negative (N) Vero cell culture at 24 hours post-infection are shown as box-and-whisker plots. The levels of P and N were compared among Delta/Adult and Omicron/Adult isolates and showed significant difference (*P* <0.05, Mann-Whitney U test).

post infection (Fig 2A). Viral proliferation level is obviously higher in cultures of Vero cells with TMPRSS2. However, the level became more similar in the absence of TMPRSS2 (Fig 2B). As assessed at 24 hours post-infection, Delta variants were produced at significantly higher levels by Vero cells in the presence of TMPRSS2 (350x) compared with its absence (Fig 2C, left). Contrary to the production of Delta variants, Omicron variants proliferated equally in TMPRSS2-positive and -negative cell cultures (9.6x) (Fig 2C, right).

The production levels of Omicron variants from adults and children were equivalent in the culture of Vero cells with and without TMPRSS2, although it was slightly superior in cultures of the cells with TMPRSS2 (Fig 3).

## Discussion

Nucleotide sequences of SARS-CoV-2 genes showed extremely high homology (more than 99.5%) with the reference strains. Although Delta variants were produced at significantly higher levels from Vero cells in the presence of TMPRSS2 *in vitro*, Omicron variants proliferated equally in TMPRSS2-postive and -negative cell cultures. Omicron variants from adults and children produced equivalent levels in the cultures of Vero cells regardless of the presence of TMPRSS2.

It has been reported that China has the lowest nucleotide diversity of SARS-CoV-2, followed by Europe and lastly by the USA and that the difference in such diversity coincides with virus transmission time order: starting in China, then Europe and finally the USA [17]. Although mutations have been accumulated over time, the diversity among circulating strains seems very low, and SARS-CoV-2 consensus sequences were reported to be indistinguishable, and they differed by 1 to 2 mutations in the rest in most of multi-infection households [18].

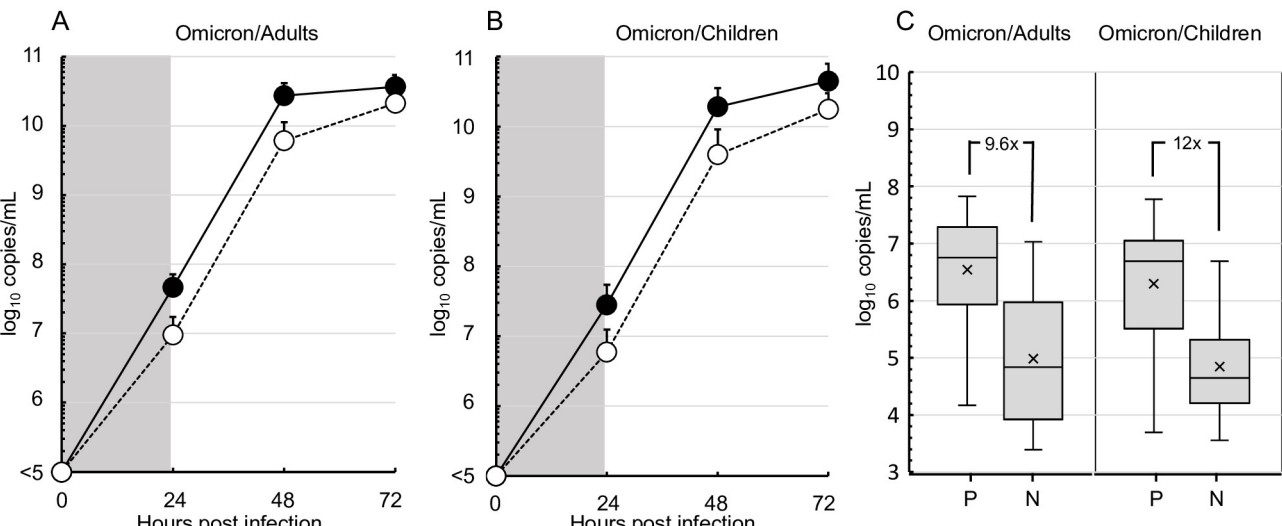

**Fig 3. Proliferation efficiency of Omicron variant clinical isolates from adult or pediatric patients in TMPRSS2-positive or -negative Vero cells.**
(A) The production levels of Omicron variants from adults (n = 3, GenBank accession numbers, LC793430-LC793432) were determined similarly in TMPRSS2 positive (●) or negative (○) Vero cell culture as shown in Fig 2. (B) The levels of Omicron variants from children (n = 3, LC793408-LC793410) were also assessed in the same manner. (C) Means and distributions of the production of Omicron variants from adults (n = 24, LC793430-LC793442, LC793444-LC793452, LC793455, LC793456) and pediatric patients (n = 21, LC793408-LC793415, LC793417-LC793429) in the supernatants of TMPRSS2-positive (P) or negative (N) Vero cell cultures at 24 hours post-infection are shown as box-and-whisker plots. No significant difference was detected between the values of P and N in omicron/adults and omicron/children (Mann-Whitney U test).

Others reported that substitutions were concentrated in the spike protein, at a rate of 0.21 amino acid residues per month [19]. Transmission efficiency depends on the unique profile of intra-host viral clones [20–22]. These findings can be attributed to the inherent nature of SARS-CoV-2 originated from its unique function of 3′-to-5′ exoribonuclease leading to high replication fidelity [23, 24]. These mutational trends were equivalent to our data shown in the phylogenetic tree from the present study. Mostly identical reference strains (>99.5% homology) were found among 100 strains computed in a Betacoronavirus BLAST search, suggesting the imported strains spread in the study area without any mutation. The profile of key amino acids in the spike protein affecting S2' cleavage was identical to the reported ones [25–27]. Therefore, the properties of the Delta and Omicron variants reported elsewhere is presumably applicable to the strains in the present study.

SARS-CoV-2 uses ACE2 for entry and the serine protease TMPRSS2 for spike protein priming [28]. In the present study, the properties of TMPRSS2 were highlighted and the degree of requirement for the TMPRSS2 molecule was assessed using clinical strains of Delta and Omicron variants. As a result, Delta variants lost the effective replication capability in the absence of TMPRSS2, but Omicron variants did not lose it. The Delta variant-specific dependence on the TMPRSS2 function was prevalent for the epidemic strains, as estimated in a model study [7]. TMPRSS2-dependent Delta variants have rarely infected children because of the lower expression of TMPRSS2 in their airway [9]. However, Omicron variants do not need enzymatic cleavage by TMPRSS2 for the replication. Independency of TMPRSS2 in Omicron variants is applicable to the strains from adults and children. Clinical isolates derived from adult and children grew equally in Vero cells regardless of the presence of TMPRSS2.

It has been proposed that Delta variants predominantly use TMPRSS2 for the cleavage at a S2' site of the SARS-CoV-2 spike protein as compared with Omicron variants, although both Delta and Omicron variants required membrane fusion for the viral replication via the fusion

peptide revealed by TMPRSS2 and/or Cathepsin L-mediated cleavage [7, 8, 28]. As expected, the predominant usage of TMPRSS2 by Delta variants was confirmed in the present study using clinical isolates from an indigenous population with similar genetic backgrounds. TMPRSS2-independent cleavage allows Omicron variants to proliferate with ease in children possessing lower levels of TMPRSS2 on epithelial surfaces of the respiratory organs [9].

This study was carried out at one site only and handled a limited number of samples. Similar studies are required at other sites to confirm the trends described in the present study. We failed in the sampling of Delta variants from pediatric patients because of rare infections caused by Delta variants in the study field. Therefore, the difference in the usage of TMPRSS2 by Delta and Omicron variants could not be examined directly using samples from pediatric patients. Even so, the above-mentioned epidemiological evidence strongly suggested that the emergence of Omicron variants and those TMPRSS2-independent entry into cells caused the sudden increase of COVID-19 epidemic size among children.

## Conclusion

The available epidemiological evidence strengthen that the TMPRSS2-dependency of SARS-CoV-2 variants is one of the most crucial determinants influencing the COVID-19 epidemic size among children.

## Supporting information

**S1 Data. This is the data for making Fig 2 panel A.**
(XLSX)

**S2 Data. This is the data for making Fig 2 panel B.**
(XLSX)

**S3 Data. This is the data for making Fig 2 panel C.**
(XLSX)

**S4 Data. This is the data for making Fig 3 panel A.**
(XLSX)

**S5 Data. This is the data for making Fig 3 panel B.**
(XLSX)

**S6 Data. This is the data for making Fig 3 panel C.**
(XLSX)

## Acknowledgments

The authors are grateful to the patients who agreed to participate in this study and to Dr. Perdana WY for technical assistance.

## Author Contributions

**Conceptualization:** Seiji Kageyama.

**Data curation:** Sosuke Kakee, Kyosuke Kanai, Akeno Tsuneki-Tokunaga, Seiji Kageyama.

**Formal analysis:** Sosuke Kakee, Kyosuke Kanai, Akeno Tsuneki-Tokunaga, Seiji Kageyama.

**Funding acquisition:** Seiji Kageyama.

**Investigation:** Sosuke Kakee, Kyosuke Kanai, Akeno Tsuneki-Tokunaga, Keisuke Okuno, Hiroki Chikumi, Seiji Kageyama.

**Methodology:** Sosuke Kakee, Kyosuke Kanai, Akeno Tsuneki-Tokunaga, Seiji Kageyama.

**Project administration:** Seiji Kageyama.

**Resources:** Noriyuki Namba, Katsuyuki Tomita, Hiroki Chikumi, Seiji Kageyama.

**Supervision:** Keisuke Okuno, Noriyuki Namba, Katsuyuki Tomita, Seiji Kageyama.

**Validation:** Sosuke Kakee, Kyosuke Kanai, Akeno Tsuneki-Tokunaga, Seiji Kageyama.

**Visualization:** Sosuke Kakee, Kyosuke Kanai, Seiji Kageyama.

**Writing – original draft:** Sosuke Kakee, Keisuke Okuno, Noriyuki Namba, Katsuyuki Tomita, Hiroki Chikumi, Seiji Kageyama.

**Writing – review & editing:** Seiji Kageyama.

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
