## [Decision Letter · Decision Letter 0]

27 Mar 2024

PONE-D-24-05165Difference in TMPRSS2 usage by Delta and Omicron variants of SARS-CoV-2: Implication for a sudden increase among childrenPLOS ONE

Dear Dr. Kageyama,

Thank you for submitting your manuscript to PLOS ONE. After careful consideration, we feel that it has merit but does not fully meet PLOS ONE’s publication criteria as it currently stands. Therefore, we invite you to submit a revised version of the manuscript that addresses the points raised during the review process.

We look forward to receiving your revised manuscript.

Kind regards,

Huseyin Tombuloglu

Academic Editor

PLOS ONE

Journal Requirements:

https://bmjpaedsopen.bmj.com/content/bmjpo/7/1/e001874.full.pdf

In your revision ensure you cite all your sources (including your own works), and quote or rephrase any duplicated text outside the methods section. Further consideration is dependent on these concerns being addressed.

   "This work was supported in part by a Grant-in-Aid for Scientific Research on Infection Control and Prevention by the International Platform for Dryland Research and Education, Tottori University."

5. We note that your Data Availability Statement is currently as follows: All relevant data are within the manuscript and its Supporting Information files.

7. Please include your tables as part of your main manuscript and remove the individual files. Please note that supplementary tables (should remain/ be uploaded) as separate "supporting information" files

8. Please provide additional details regarding participant consent. In the ethics statement in the Methods and online submission information, please ensure that you have specified what type you obtained (for instance, written or verbal, and if verbal, how it was documented and witnessed). If your study included minors, state whether you obtained consent from parents or guardians. If the need for consent was waived by the ethics committee, please include this information.

Reviewers' comments:

Reviewer's Responses to Questions

**Comments to the Author**

1. Is the manuscript technically sound, and do the data support the conclusions?

Reviewer #1: Yes

Reviewer #2: Yes

2. Has the statistical analysis been performed appropriately and rigorously? 

Reviewer #1: Yes

Reviewer #2: Yes

3. Have the authors made all data underlying the findings in their manuscript fully available?

Reviewer #1: No

Reviewer #2: Yes

4. Is the manuscript presented in an intelligible fashion and written in standard English?

Reviewer #1: Yes

Reviewer #2: Yes

5. Review Comments to the Author

Reviewer #1: Analysis of indicated groups depending specific conditions are valuable for comparing the spreading rate of the Covid-19. The article has some disadvantages such as limited number of samples and population as indicated by the authors. But results of obtained samples support the hypothesis.

Additionally;

-In article, Table 1 that has been indicated in lane 194 and 200 was missing.

-Word ‘positive’ has been written mistakenly as ‘postive’ in some part of the article.

-Discussion and conclusion part would be written in more detail and strong.

Reviewer #2: The study entitled 'Difference in TMPRSS2 usage by Delta and Omicron variants of SARS-CoV-2:

Implication for a sudden increase among children' is descriptive in nature and presents significant findings. With regard of the study "Analysis of epidemic strains from indigenous populations with similar genetic backgrounds 275 in one area showed that the Omicron variants were able to replicate independently in 276 pediatric patients with lower levels of TMPRSS2 on epithelial surface of the respiratory". It is acceptable.

6. PLOS authors have the option to publish the peer review history of their article (what does this mean?). If published, this will include your full peer review and any attached files.

Reviewer #1: No

Reviewer #2: No

---

## [Author Response · Author response to Decision Letter 0]

30 Apr 2024

Dr. Huseyin Tombuloglu

Academic Editor

PLOS ONE

---

## [Editor Report · Decision Letter 1]

7 May 2024

Difference in TMPRSS2 usage by Delta and Omicron variants of SARS-CoV-2: Implication for a sudden increase among children

PONE-D-24-05165R1

Dear Dr. Kageyama,

We’re pleased to inform you that your manuscript has been judged scientifically suitable for publication and will be formally accepted for publication once it meets all outstanding technical requirements.

Kind regards,

Huseyin Tombuloglu

Academic Editor

PLOS ONE
---

## [Editor Report · Acceptance letter]

13 May 2024

PONE-D-24-05165R1 

PLOS ONE

Dear Dr. Kageyama, 

I'm pleased to inform you that your manuscript has been deemed suitable for publication in PLOS ONE. Congratulations! Your manuscript is now being handed over to our production team.

Kind regards, 

on behalf of

Dr. Huseyin Tombuloglu 

Academic Editor

PLOS ONE